# Vitamin D Supplementation in Patients with Juvenile Idiopathic Arthritis

**DOI:** 10.3390/nu14081538

**Published:** 2022-04-07

**Authors:** Chao-Yi Wu, Huang-Yu Yang, Shue-Fen Luo, Jing-Long Huang, Jenn-Haung Lai

**Affiliations:** 1Division of Allergy, Asthma and Rheumatology, Department of Pediatrics, Chang Gung Memorial Hospital, Taoyuan 333, Taiwan; joywucgu@hotmail.com (C.-Y.W.); hjlong0182@gmail.com (J.-L.H.); 2College of Medicine, Chang Gung University, Taoyuan 333, Taiwan; hyyang01@gmail.com; 3Department of Nephrology, Chang Gung Memorial Hospital, Taoyuan 333, Taiwan; 4Division of Allergy, Immunology and Rheumatology, Department of Internal Medicine, Chang Gung Memorial Hospital, Chang Gung University, Taoyuan 333, Taiwan; lsf00076@adm.cgmh.org.tw; 5Department of Pediatrics, New Taipei Municipal TuCheng Hospital, New Taipei City 236, Taiwan; 6National Defense Medical Center, Graduate Institute of Medical Science, Taipei 114, Taiwan

**Keywords:** juvenile idiopathic arthritis, vitamin D, supplementation, immune regulation, bone metabolism

## Abstract

Vitamin D has been implicated in the pathogenesis of skeletal disorders and various autoimmune disorders. Vitamin D can be consumed from the diet or synthesized in the skin upon ultraviolet exposure and hydroxylation in the liver and kidneys. In its bioactive form, vitamin D exerts a potent immunomodulatory effect and is important for bone health. Juvenile idiopathic arthritis (JIA) is a collection of inflammatory joint diseases in children that share the manifestation of inflamed synovium, which can result in growth arrest, articular deformity, bone density loss, and disability. To evaluate the potential effect of vitamin D on JIA disease manifestations and outcomes, we review the role of vitamin D in bone metabolism, discuss the mechanism of vitamin D in modulating the innate and adaptive immune systems, evaluate the clinical significance of vitamin D in patients with JIA, and summarize the supplementation studies.

## 1. Introduction

Juvenile idiopathic arthritis (JIA) is the most prevalent debilitating rheumatic disease of childhood, with a prevalence rate of 1/1000 worldwide [1]. JIA comprises heterogeneous subtypes of diseases with complex immunopathology and leads to a common presentation, namely the inflammation and thickening of the joint lining, with characteristic onset in children before the age of 16 [2]. The immune cells and the inflammatory mediators interleukin (IL)-1, IL-6, and IL-17, and the tumor necrosis factor (TNF)-α are well-studied cytokines that actively contribute to the clinical phenotypes of JIA, such as synovial inflammation and bone resorption [3,4]. However, despite the success of conventional and new biological disease-modifying antirheumatic drugs (DMARDs) with cytokine-blocking therapeutic approaches, a substantial percentage of patients will have ongoing active disease into adulthood that includes sequelae from chronic inflammation and considerable morbidity from joint damage, osteoporosis, growth retardation, uveitis, and cardiovascular problems, which can significantly impact patient quality of life [5]. Accumulating evidence suggests that the presence and activity of JIA may be associated with a reduced concentration of serum vitamin D [6,7].

Vitamin D was first known as an essential element in modulating bone health and altering calcium homeostasis [8,9]. Vitamin D deficiency can result in osteomalacia in adults and rickets in children [8,9]. Beyond its endocrine activity, its immunoregulatory property of being capable of enhancing the immunomodulatory activities of monocytes and macrophages, and the downregulating of the proinflammatory cytokines produced by numerous T lymphocytes has been discussed extensively [10,11,12]. Considerable literature has reviewed the links between vitamin D deficiency and various infectious conditions as well as autoimmune diseases, including inflammatory bowel disease, type 1 diabetes, multiple sclerosis, rheumatoid arthritis (RA), systemic sclerosis, and systemic lupus erythematosus [13,14,15,16,17,18,19,20,21]. Following the discovery of vitamin D receptors (VDRs) in cells such as monocytes, macrophages, dendritic cells, and lymphocytes, the molecular basis for the regulatory effect of vitamin D on immune cells and the potential benefits of vitamin D supplementation in autoimmune diseases have gradually been revealed [15,16,17]. In the present review, we discuss the effect of vitamin D on bone metabolism and carefully summarize how it modulates the immune system. In addition, we evaluate the clinical significance of vitamin D in patients with JIA and summarize the available supplementation studies of vitamin D.

## 2. Vitamin D

Vitamin D is a collection of secosteroids primarily synthesized upon the catalysis of ultraviolet B (UVB) radiation. Among them, vitamin D_2_ (also known as ergocalciferol) and vitamin D_3_ (also known as cholecalciferol) are perhaps the most discussed vitamin D compounds [22]. Vitamin D_2_ is mostly generated in yeast and mushrooms upon UVB irradiation. Vitamin D_3_, on the other hand, is formed from UVB-irradiated 7-dehydrocholesterol in the epidermal skin layer and can also be obtained from diets to a much lesser extent [23]. Neither synthesized nor ingested vitamin D, however, is biologically active. The activation of vitamin D via enzyme hydroxylation in the host liver and kidneys is essential for its endocrinal activity as well as immunomodulatory effects. Specifically, extracellular vitamin D is transported to the liver, where it is hydroxylated to 25-hydroxyvitamin D (25[OH]D; also known as calcidiol) by 25-hydroxylase (CYP2R1) [24]. 25[OH]D, the major circulating form of vitamin D, is then transported to the proximal tubule cells within the kidneys and further hydroxylated to its biologically active form, namely 1,25 dihydroxyvitamin D (1,25[OH]_2_D; also known as calcitriol). This process is catabolized by 1α-hydroxylase (CYP27B1) under the regulation of the parathyroid hormone (PTH), calcium, phosphate, and fibroblast growth factor 23 (FGF23) [24]. However, kidneys are not the only source of 1α-hydroxylase. Extrarenal synthesis of 1,25[OH]_2_D is performed in various immune cells and tissues, including the skin, brain, colon, and pancreas, especially under the stimulation of interferon (IFN)-γ, IL-1β, and INF-α [24,25]. Circulating vitamin D and its metabolites require the binding of the vitamin D-binding protein (DBP) for transportation. Notably, DBP, aside from its escorting role, also regulates bone development and contributes to immune modulation [22,26].

1,25[OH]_2_D binds to the intracellular VDR, a transcription factor, to exert its genomic effects. Upon the binding of 1,25[OH]_2_D, VDR dimerizes with other nuclear receptors and binds to vitamin D response elements (VDREs) in the promotor regions to regulate the expression of genes mediating various biological activities [24,25]. Moreover, the non-genomic effects of vitamin D are also discovered to proceed through caveolin-1, the complex of 1,25[OH]_2_D with membrane-bound VDR [27]. To avoid overwhelming vitamin D signaling and potential toxicity, negative feedback of 1,25[OH]_2_D-induced 24-hydroxylase (CYP24A1) further catabolizes 25[OH]D and 1,25[OH]_2_D to 24,25[OH]_2_D and 1,24,25[OH]_3_D in the kidneys [24,25]. The major steps in vitamin D metabolism are demonstrated in Figure 1.

The half-life of biologically active 1,25[OH]_2_D is approximately 5–8 h in humans [28]. Due to its rapid turnover, levels of 25[OH]D (1,25[OH]_2_D precursor) are mostly used to determine the overall vitamin D status and clinically relevant vitamin D reserve [29,30,31]. Although no consensus has been made on the cutoff level of vitamin D in overall physical and pathogenic conditions, it is widely accepted that a minimum serum concentration of 75 nmol/L (30 ng/mL) is the lower limit of the optimal vitamin D level due to its associated reduced risk for bone fracture [32]. Nonetheless, other guidelines recommend serum 25[OH]D concentrations of >50 nmol/L (20 ng/mL) for optimal bone health in older adults [29,30,31]. Supplementary corrections are usually required in individuals with severe vitamin D deficiency with a serum 25[OH]D concentration of <30 nmol/L (12 ng/mL) [29,30].

## 3. Role of Vitamin D in Bone Metabolism

Longitudinal bone growth via endochondral bone formation requires a sufficient size of the proliferative pool, adequate matrix synthesis, and hypertrophic chondrocyte enlargement [33]. The mineralization and calcification of the collagenous matrix requires vascular invasion from the marrow of the metaphysis, providing a channel for the recruitment of osteoclasts and differentiating osteoblasts that remodel, ossify, and cartilage tissues into hard bones [33]. As bone cells, including chondrocytes, osteoblasts, osteocytes, and osteoclasts, all express VDR [8,34] and 1,25[OH]_2_D can directly affect bone homeostasis in response to various inflammatory mediators in patients with JIA [33]. Moreover, vitamin D is classically known for its role in regulating calcium homeostasis and maintaining skeletal integrity [8,33,34]. Considering that exaggerated growth and active skeletal mineralization occur during the time of childhood and adolescence, any disruption underlying modulatory variables, such as inflammatory mediators and inadequate levels of vitamin D, may alter skeletal development in patients with JIA [33].

### 3.1. Indirect Effect of Vitamin D on Bone Homeostasis

Indirectly, 1,25[OH]_2_D serves as a potent stimulator for VDR-mediated intestinal calcium absorption to increase the calcium pool required for proper bone mineralization [8]. When the level of 1,25[OH]_2_D is low, absorption of calcium in the intestines is decreased, resulting in a reduced level of serum calcium. To preserve serum calcium, the parathyroid glands actively produce more PTH, increasing renal reabsorption and moving calcium out of the bones. In addition, PTH also stimulates the renal synthesis of CYP27B1 to promote the hydroxylation of 25[OH]D into 1,25[OH]_2_D [8,34]. As a negative feedback loop, however, the elevation of 1,25[OH]_2_D inhibits PTH secretion, suppresses CYP27B1, and stimulates CYP24A1 in the kidney to enhance its catabolism [8,34].

### 3.2. Direct Effect of Vitamin D on Bone Homeostasis

Vitamin D also directly affects bone homeostasis via VDR binding on bone cells. Specifically, 1,25[OH]_2_D and 24,25[OH]_2_D are both important for proper endochondral bone formation [8]. VDR signaling in chondrocytes alters bone mass via the receptor activator of nuclear factor kappa-Β ligand (RANKL) secretion and the induction of osteoclasts [35]. VDR signaling also regulates the production of FGF23 in osteoblasts, which consequently leads to increased renal CYP27B1 expression and renal phosphate reabsorption [35]. By enhancing osteoblast membrane-associated RANKL-RANK osteoclast stimulation, 1,25[OH]_2_D promotes osteoclastogenesis and bone resorption, in addition to its known role in promoting skeletal mineralization [8,34]. In addition, 1,25[OH]_2_D also directly inhibits skeletal mineralization by enhancing the expression of mineralization inhibitors during a negative calcium balance [34]. Interestingly, VDR signaling does not seem to affect bone homeostasis under normal serum calcium concentrations in osteogenic cells [34]. Nonetheless, although VDR is located on both osteoblasts and osteoclasts, the results from VDR knockout mice suggested that osteoblasts are the key responders to 1,25[OH]_2_D-induced osteoclastogenesis [8,34].

As depicted in Figure 2, the sum effect of 1,25[OH]_2_D-VDR signaling in osteogenic cells, intestines, and kidneys mainly contributes to the balance of serum calcium by reducing mineralized bone mass when intestinal calcium absorption is decreased [34].

## 4. Immune Modulatory Role of Vitamin D

Monocytes, macrophages, and dendritic cells are critical immune cells regulating the secretion of JIA-associated pathogenic proinflammatory mediators, including IL-1, IL-6, and IL-18 S100 proteins [3,4]. In addition, the imbalance of T helper-1 (Th1)/T helper-17 (Th17) cells and regulatory T cells (Tregs) leads to the loss of immune tolerance against self-antigens and further contributes to the inflammation of articular synovium and loss of bone mineralization in oligo/polyarticular JIA [3]. To understand the potential impact of vitamin D on altering the inflammatory status in patients with JIA, we briefly review the effects of VDR signaling within immune cells (Figure 3).

### 4.1. Innate Immune System

Monocytes and macrophages are important players in the innate immune system, orchestrating the initiation, progression, and resolution of many autoimmune diseases [36]. Upon IFN-γ and Toll-like receptor (TLR) signaling, the expression of CYP27B1, capable of catabolizing 25[OH]D to 1,25[OH]_2_D, in monocytes and macrophages is upregulated [24,37,38]. While monocytes, as macrophages, express VDR, the level of VDR expression is reduced as monocytes differentiate toward macrophages [39]. Through VDR signaling, 1,25 [OH]_2_D stimulates antimicrobial peptide transcription, alters the formation of oxygen radicals, increases IL-10 secretion, and modulates the production of proinflammatory cytokines in monocytes and macrophages [22,40,41,42,43,44,45,46,47]. Recently, 1,25[OH]_2_D was shown to modulate the epigenome among immune cells, particularly monocytes, while encountering antigens and during innate immune system differentiation [48,49].

Dendritic cells bridge the innate and adaptive immune systems, and play a critical role in directing the development of lymphocytes in various autoimmune disorders [50]. Similar to monocytes and macrophages, dendritic cells also express VDR and CYP27B1. Through VDR signaling, 1,25[OH]_2_D modulates the innate–adaptive immune interface, resulting in a less inflammatory dendritic cell phenotype [12,14,43,51,52]. Specifically, 1,25[OH]_2_D directs dendritic cells toward a more tolerogenic and less mature phenotype by decreasing the surface major histocompatibility complex (MHC) II, costimulatory molecules, and adhesion molecules while increasing the expression of chemokine receptors, antigen-uptake receptors, and macrophage markers [53,54,55]. Additionally, dendritic cells also increase the secretion of IL-10 and reduce the production of IL-6, IL-12, and IL-23 upon VDR signaling [22,53,54,55]. Moreover, 1,25[OH]_2_D modulates dendritic cells to suppress autoreactive T lymphocyte proliferation, to promote apoptosis in T lymphocytes, and to induce Tregs through the expression of programmed death-ligand 1 (PD-L1) and TNF secretion [56,57,58].

Although neutrophils lack CYP27B1, its abundance of VDR mRNA expression is comparable with monocytes [15,59]. Recent study suggested that vitamin D enhances the formation of neutrophil extracellular traps (NETs), upregulates the production of anti-inflammatory cytokine IL-4, and downregulates proinflammatory cytokines IL-1β, IL-6, IL-8, and IL-12 in neutrophils under various infectious conditions [15,60]. Interestingly, while vitamin D upregulates the pattern recognition receptors TLR2 and nucleotide binding oligomerization domain containing 2 (NOD2), it also negatively regulates TLR-induced inflammation via the suppression of cytokine signaling (SOCS)-1 and SOCS-3 induction in an IL-4-dependent manner [60]. Evidence also suggested that among chronic obstructive pulmonary disease patients treated with 1,25(OH)2D, there is an increase of apoptosis in neutrophils [61]. Moreover, vitamin D downregulates IL-15 production and suppresses immunoglobulin-E (IgE)-dependent mast cell activation via eosinophil alteration [62,63].

### 4.2. Adaptive Immune System

The antigen-specific immune response is mediated by the interaction of lymphocytes and antigen-presenting dendritic cells. According to the cytokine profile expressed by dendritic cells upon VDR signaling as described above, 1,25[OH]_2_D is known to induce Treg while suppressing Th1 and Th17 differentiation [22,53,54,55,56,57,58]. Moreover, following the discovery of CYP27B1 and VDR in T lymphocytes, the direct effects of 1,25[OH]_2_D on T cells were also revealed [64,65]. Specifically, during the activation of naive T cells, the expression of VDR gradually increases [64,65]. 1,25[OH]_2_D directs the differentiation of CD4 T cells, suppressing the production of the Th1 and Th17 cytokines, and promoting Th2 differentiation by inhibiting IL-2, IL-17, IL-21, IL-22, IFN-γ, and TNF-α secretion as well as promoting IL-4, IL-5, and IL-10 production [12,65,66]. Furthermore, 1,25[OH]_2_D limits proinflammatory immune responses by inducing the differentiation of Tregs via IL-10 and FoxP3 expression [65,67]. Recent study further confirmed the importance of VDR as a predicted key regulator in the synovial fluid effector Treg cells, positively correlated with both FoxP3 and T-bet expression among patients with JIA [68].

Similar to T cells, B lymphocytes are also positive for CYP27B1 and VDR. 1,25[OH]_2_D has been shown to interfere with the production of antibodies and induce apoptosis among activated B cells [26]. Additionally, 1,25[OH]_2_D suppresses the differentiation of naive B cells and their maturation [69]. Due to the production of IL-10 and the decreased expression of surface CD86 on B cells upon VDR signaling, B lymphocytes are also capable of suppressing T-cell activation [70,71].

### 4.3. Others

Synovial fibroblasts overlying the joint space express VDR and are also capable of reacting to vitamin D cues [72,73,74,75]. Although the direct effect of vitamin D on JIA synovial fibroblasts has not yet been investigated, the presence of suboptimal levels of vitamin D reduces the expression of TNF-α and IL-6, and upregulates the osteoprotegerin (OPG)/RANKL ratio in RA fibroblast-like synoviocytes, inhibiting inflammation-induced osteoclastogenesis [73]. Moreover, high levels of 1,25[OH]_2_D have been shown to inhibit the invasion of synovial fibroblasts and suppress the production of both IL-1β and matrix metalloprotease-1 [74]. As 1,25[OH]_2_D has been suggested to negatively regulate inflammatory responses only during inflammatory status [76], the apoptosis of synovial fibroblasts induced by 1,25[OH]_2_D is a great example as it only occurs in the presence of TNF-α [75].

## 5. Clinical Significance of Vitamin D in Patients with JIA

### 5.1. Vitamin D Level and the Presence of JIA

Serum 25[OH]D was reported to be deficient or insufficient in a high number of JIA patients across numerous studies [6,7,77,78,79,80,81,82]. Even though few data suggested otherwise [83,84,85], a suboptimal level of 25[OH]D among JIA patients was reported in up to 84.2% of the JIA studies (32 out of 38) [7]. In addition, vitamin D was considered to be insufficient in up to 82% of JIA patients in a meta-analysis study [82]. In a recent review, 64.3% (nine out of 14) of the JIA studies reported a significantly lower mean 25[OH]D concentration among JIA patients than among their healthy counterparts [7]. Interestingly, to understand whether individuals with suboptimal concentrations of vitamin D are at higher risk of developing JIA, Thorsen et al. conducted a case–control study measuring the level of vitamin D at birth [86]. His work revealed no significant association between the levels of 25[OH]D at birth and later risk of JIA [86]. Recently, utilizing the Mendelian randomization method, Clarke et al. found no causal relationship between 25[OH]D levels and JIA [87]. According to his analysis, vitamin D supplementation at the population level is unlikely to reduce JIA incidence [87].

### 5.2. Vitamin D Level and BMD in Patients with JIA

Considering the role of vitamin D in bone metabolism, the association of the 25[OH]D level with bone mineral density (BMD) in patients with JIA has also been investigated [88,89,90,91]. Remarkably, in a cross-sectional study containing 48 JIA patients, the level of serum 1,25[OH]_2_D concentration was shown to accurately differentiate 79.6% of the patients into either low or normal-BMD groups [88]. A marginally significant correlation between trabecular bone density and 25[OH]D concentrations was also reported [91]. Based on the discovery of reduced 25[OH]D values and higher PTH levels among JIA patients, particularly those with the systemic onset type [78,89], it is speculated that vitamin D and its active metabolites may in part explain why JIA patients do not achieve normal bone mineralization over time despite the use of more effective drugs [78]. Nonetheless, as the degree of inflammation and the use of systemic steroids can both significantly influence bone mineralization [92,93], the direct impact of vitamin D on bone metabolism among patients with JIA should be interpreted with caution.

### 5.3. Vitamin D Level and Disease Activity of JIA

Interest in how vitamin D levels alter JIA disease activity has grown along with our understanding of its ability to modulate inflammation [6,7,80,81,85,94]. Controversies, however, were shown in a recent review [7]. Approximately half of the recruited studies (seven out of 15) showed an inverse correlation of 25[OH]D levels among patients with higher disease activity or elevated inflammatory biomarkers [7]. Although the opposite relationship was demonstrated in one study, the remaining half of the studies also revealed an inverse association of vitamin D and disease activity without statistical significance [7]. Recent studies regarding the influence of vitamin D on JIA disease activity revealed a similar trend [6,81,85,94,95]. Moreover, a decreased level of serum 25[OH]D was also reported to correlate with the presence of JIA-associated uveitis and JIA disease duration [6,81]. Considering real-world epidemiological studies and the underlying immune modulating mechanisms, a potential benefit arises in correcting vitamin D deficiency in patients with JIA with adequate vitamin D supplementation.

### 5.4. Genomic Association of Vitamin D Signaling and JIA

The significance of vitamin D in JIA was also documented at the genomic level. An increased utilization of vitamin D during active inflammation, possibly operated by VDR polymorphisms, was theorized in patients with autoimmune diseases [96,97]. Among the 100 kb VDR gene located on human chromosome 12q13.1, *ApaI*, *BsmI*, *Cdx2*, *FokI*, and *TaqI* are the most widely studied polymorphisms. While vitamin D status was independent of *VDR* genotypes, polymorphisms of *ApaI* and *Cdx2* were significantly different between JIA patients and unaffected controls [7,98,99]. *Cdx2* polymorphism is located in the VDR gene promotor region associating with gene transcription in the intestinal epithelial cells and influences intestinal calcium absorption [99]. Although *ApaI* polymorphism is not functional itself, polymorphisms of *ApaI* have been hypothesized to modulate VDR mRNA stability or be in linkage disequilibrium with other adjacent gene loci exerting a role in JIA pathogenesis [98]. Interestingly, polymorphisms of *VDR* have also been found to alter the lipid profile among JIA patients in Serbia [100]. Recently, the gene–gene interaction between *PTPN2* and vitamin D pathway genes was demonstrated and replicated in two independent JIA case–control cohorts [101]. Specifically, the minor allele of the vitamin D pathway genes, including *GC*, *CYP24A1*, *CYP2R1*, and *DHCR7*, were associated with both reduced binding protein concentrations and reduced levels of circulating 25[OH]D [101]. As *PTPN2* gene expression is known to be regulated by vitamin D, the observed gene–gene interactions indicated epistasis amongst *PTPN2* and the genes of the vitamin D pathway in contributing to the risk of JIA [101].

## 6. Vitamin D Supplementation in JIA

Hypovitaminosis D is currently a global issue for children worldwide and an intake of 400 IU and 600 IU vitamin D daily is generally accepted to prevent nutritional rickets in infants and children [31,102]. While limited foods such as fortified milk and oily fish contain meaningful amounts of vitamin D, sunlight exposure is usually essential for adequate vitamin D synthesis. Factors including skin pigmentation, latitude, season, the time of day, and exposure surface area can all influence the efficiency of vitamin D production resulting from sunlight exposure. Considering all variables, “sensible sun exposure” with sun exposure on the arms and legs for 5–30 min between 10 a.m. and 3 p.m. twice weekly was recommended to maintain vitamin D concentrations and avoid deficiency [103]. The American Academy of Dermatology further stated that the maximum production of vitamin D occurs after 2 to 5 min of midday summer exposure for a light-skinned person living in New York [104]. Due to the risk of skin cancer, however, deliberate and unprotected sun exposure to correct vitamin D insufficiency is not recommended. The supplementation of exogenically synthesized vitamin D with ultraviolet-treated yeast ergosterol and lanolin 7-dehydrocholesterol has now become relatively common.

Considering the high prevalence of hypovitaminosis and the usage of steroids in patients with JIA, special recommendations on the optimal vitamin D requirements for children with chronic inflammatory rheumatic conditions have been discussed in the past decade [105,106]. Von Scheven and Burnham recommended a 400 IU daily regimen as the minimum dosing supplementation, as suggested by the American Academy of Pediatrics guidelines [105]. Although vitamin D has an outstanding safety profile [29,107], considering the medicinal effects of additional nutritional supplements, the authors suggested a cautious approach and the need for further studies to clarify the potential benefits and risks of vitamin D supplementation among children with rheumatic conditions [105]. Recently, Vojinovic and Cimaz suggested the guidelines established by the Endocrinology Society, supplying a dose of vitamin D two to three times the current recommendation in all children with rheumatic diseases receiving systemic steroids [106].

As the epidemiologic findings and the potential benefits of vitamin D supplementation in JIA continue to emerge, only a limited prospective controlled trial study with an “optimal amount” of vitamin D supplementation has yet been reported to date [90,108,109,110,111]. In an attempt to increase BMD among JIA patients who had osteopenia, Reed et al. studied the effect of 25[OH]D supplementation (1–2 μg/kg/day for 12 months) in 13 children with sustained active polyarticular JIA [110]. They demonstrated that although there is a significant improvement in the mean 25[OH]D level, the level of 1,25[OH]_2_D, JIA disease activity, and BMD remained the same [110]. Warady et al. later prescribed Vit D with calcium supplementation for children with osteoporosis and rheumatic disease under the treatment with systemic corticosteroid [109]. A significant increase of spinal BMD (11% over baseline) was observed among the 10 enrolled patients, including six JIA, during the period of intervention [109]. However, the mean BMD decreased soon after Vit D and calcium supplementation were terminated [109]. In a supplementation study for patients with oligo or polyarticular JIA, 18 children were randomly assigned to receive either Vit D 3 (2000 IU/day) or calcium (1000 mg/day), or both for a total of 6 months [90]. Hillman et al. discovered that under Vit D3 supplementation (either alone or with calcium), the level of serum 25[OH]D increases but not 1,25[OH]_2_D [90]. No raise in the bone mineral content or changes in the level of bone turnover markers were seen upon treatment [90]. Moreover, Vit D (400 IU/day) supplementation with or without calcium (1000 mg/day) for 24 months was investigated in a study including 198 patients with JIA [111]. While additional calcium supplementation accelerated the rise of BMD, a linear increase of average total BMD over time was documented in both the vitamin D3 plus calcium and the vitamin D3-only group [111]. Finally, in the most recent randomized controlled trail study, the supplementation of vitamin D_3_ at a dose of 2000 IU/day for 24 weeks did not significantly reduce disease activity or improve BMD patients with JIA despite the rise of serum 25[OH]D [108]. A summary of the regimens and effect of vitamin D supplementation in JIA are carefully reviewed and displayed in Table 1 below.

Several studies calculated the estimated amount of daily vitamin D intake and reported the supplementation of vitamin D among JIA patients in real-world practice. Many of the vitamin D dosing, however, did not meet the recommended minimal amount of 400 IU daily. Utilizing a three-day diet recall method, Dey et al. calculated daily calcium and vitamin D intake among 35 patients with JIA and their age/gender-matched controls. Lower daily vitamin D intake (123 ± 53.6 IU) was reported among the patients as compared to the controls (309 ± 62.38 IU) [112]. While Pepmueller et al. reported an intake of 464 ± 262 IU vitamin D among patients with JIA, data from Stagi and Lien suggested a much lower load of vitamin D (around 140–165 IU/day) intake in JIA patients, which is no different from their healthy counterparts [78,113,114]. Around 32–52% of the JIA patients were prescribed with daily supplementation of vitamin D 400–800 IU in Helsinki, Finland [115,116].

## 7. Concluding Remarks

JIA is the leading rheumatic disorder driven by a dysregulated innate and adaptive immune system in children. While current treatments provide notable improvements in the control of systemic inflammation and the relieve of symptoms in JIA, issues in long-term bone health and concerns for medication-related side effects, infection, and malignancy, for example, have led to the search for a cost-effective therapy or nutrient supplementation with limited side effects. Vitamin D is an essential hormone in altering calcium homeostasis and modulating bone health. Its deficiency can result in osteomalacia and rickets in humans [8,9]. Beyond its endocrine activity, vitamin D is capable of enhancing the immunomodulatory activities of monocytes and macrophages, and suppressing the proinflammatory cytokines produced by lymphocytes as summarized above [10,11,12]. Considering the prevalence of vitamin D insufficiency among patients with JIA and its ability to alter bone mineralization and immune modulation, rising interests have led to the research of vitamin D supplementation in JIA.

Clinical significance of vitamin D in patients with JIA has been documented. Data from limited randomized controlled trials, however, did not support the therapeutic effect of vitamin D supplementation in suppressing disease activity or improving bone mineralization. Heterogeneity between studies in terms of the study population, vitamin D supplementation regimen, outcome measurements, and definitions of vitamin D deficiency is perhaps the most obvious problem. Moreover, as trials were conducted across 30 years in different geographic locations, with various ethnic backgrounds, under diverse qualities of diet, sun exposure, and clothing style, it is hard to draw a conclusion based on the available information at present.

Further studies are needed to better evaluate the potential efficacy of vitamin D treatment in JIA. First, larger controlled trials are recommended as sample sizes from existing JIA supplementation studies were small. These trials do not allow for stratification based on baseline vitamin D concentration, JIA subtypes, disease activity, medication, diet quality, sun exposure, and other factors. Next, given the high prevalence of vitamin D deficiency and because vitamin D is noted to be safe for adults in doses of up to 4000 IU/day [117], trials should supply sufficient vitamin D to correct deficiencies.

In conclusion, mechanistic and epidemiologic data support a role for vitamin D in promoting bone health and modulating disease status in JIA. However, additional evidence is needed to clarify the efficacy of vitamin D supplementation in patients with JIA and determine the optimal level of vitamin D for these patients.

## Figures and Tables

**Figure 1 nutrients-14-01538-f001:**
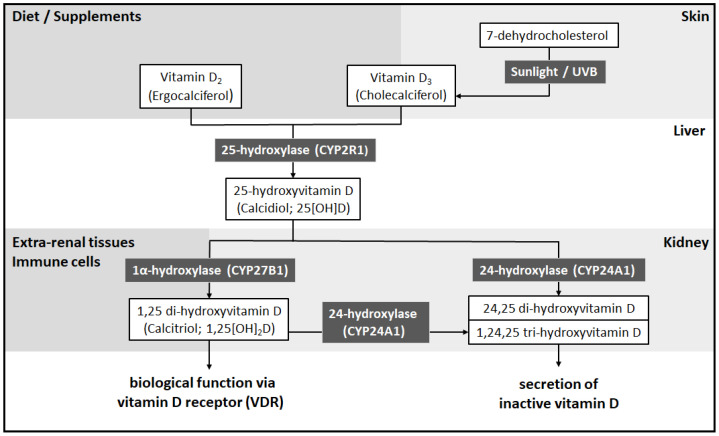
Metabolism of vitamin D. Vitamin D can be either produced in the skin upon UVB radiation or ingested through dietary intake. Vitamin D (representing vitamin D_2_ and D_3_) is converted by 25-hydroxylase (CYP2R1) in the liver to 25[OH]D. 25[OH]D is converted in the kidneys and several extrarenal tissues, including immune cells, by 1α-hydroxylase (CYP27B1) to 1,25[OH]_2_D. 1,25[OH]_2_D, the biologically active form of vitamin D, binds to the vitamin D receptor to carry out its biological function. To avoid overwhelming vitamin D stimulation, 24-hydroxylase (CYP24A1) further catabolizes 25[OH]D and 1,25[OH]_2_D to 24,25 dihydroxyvitamin D (24,25[OH]_2_D) and 1,24,25[OH]_3_D for excretion in the kidneys. Notably, although not depicted in the present figure, the binding of vitamin D and its metabolites to DBP is required for their transportation in the circulation. Abbreviations: 1,25[OH]_2_D—1,25 dihydroxyvitamin D; 25[OH]D—25-hydroxyvitamin D; 24,25[OH]_2_D—24,25 dihydroxyvitamin D; 1,24,25[OH]_2_D—1,24,25 tri-hydroxyvitamin D; and D-binding protein—DBP.

**Figure 2 nutrients-14-01538-f002:**
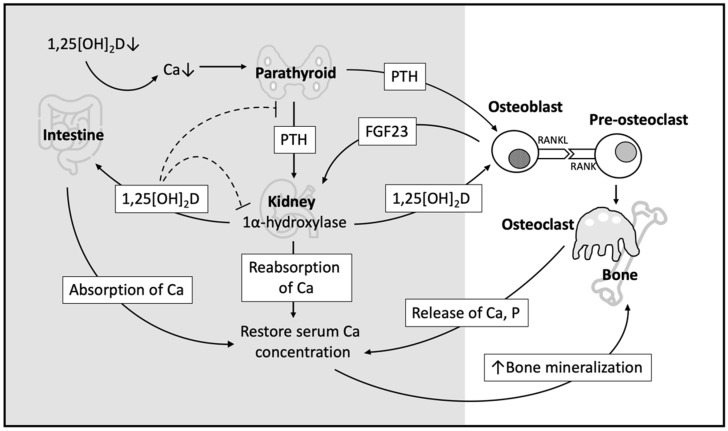
Role of vitamin D in bone metabolism. A low level of 1,25[OH]_2_D decreases calcium absorption in the intestines and results in a low level of serum Ca. The hypothyroid gland senses changes in serum Ca and secretes PTH when the Ca level is reduced. PTH stimulates the expression of 1α-hydroxylase and promotes the production of 1,25[OH]_2_D in kidneys. 1,25[OH]_2_D enhances intestinal absorption of Ca and stimulates the expression of RANKL on osteoblasts to interact with RANK on preosteoclasts to perform mature osteoclastic activity, releasing calcium and phosphorus from the bone. The reabsorption of Ca in the kidneys further restores the level of serum Ca. Additionally, 1,25[OH]_2_D inhibits 1α-hydroxylase and limits parathyroid glands from secreting PTH. In summary, the effect of 1,25[OH]_2_D-VDR signaling in osteogenic cells, intestines, and kidneys mainly contributes to the balance of serum calcium when intestinal calcium absorption is decreased. Abbreviations: 1,25[OH]_2_D—1,25 dihydroxyvitamin D; Ca—calcium; P—phosphorus; PTH—parathyroid hormone; RANKL—nuclear factor kappa-Β ligand; RANK—nuclear factor kappa-Β; VDR—vitamin D receptor; and FGF23—fibroblast growth factor 23.

**Figure 3 nutrients-14-01538-f003:**
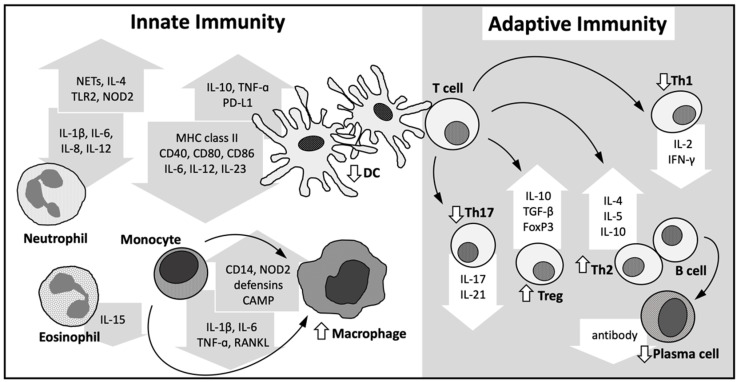
The impact of vitamin D on immune cells. Vitamin D stimulates macrophage differentiation, induces the production of proinflammatory cytokines, and promotes antimicrobial peptide expression as well as PRP in monocytes and macrophages. It decreases the differentiation, maturation, and antigen presentation ability of dendritic cells while decreasing the production of IL-6, IL-12, and IL-23. Vitamin D induces formation of NETs and PRP expression but suppresses proinflammatory cytokines in neutrophils. It promotes the release of IL-4 from neutrophils yet decreases eosinophil IL-15 production. An immunomodulatory role of vitamin D in dendritic cells is also observed through the increase in IL-10 secretion and the induction of T-cell apoptosis via the expression of PD-L1 and TNF in Tregs. By moving helper T cells toward a more tolerogenic state, vitamin D directly induces Th2 and Treg differentiation, and downregulates proinflammatory Th1 and Th17 lymphocytes. Additionally, mediated via vitamin D-VDR signaling, decreased proliferation of B cells and its differentiation to plasma cells can lead to low antibody production. Abbreviations: PRP—pattern recognition pattern; IL—interleukin; PD-L1—programmed death-ligand 1; TNF—tumor necrosis factor; Treg—regulatory T cell; Th—helper T cell; VDR—vitamin D receptor; 1,25[OH]_2_D—1,25 dihydroxyvitamin D; Ca—calcium; P—phosphorus; PTH—parathyroid hormone; VDR—vitamin D receptor; MHC—major histocompatibility complex; NOD2—nucleotide binding oligomerization domain containing 2; CAMP—cathelicidin antimicrobial peptide; TGF—transforming growth factor; IFN– interferon; FoxP3—forkhead box P3; TLR2—toll like receptor 2; and NETs—neutrophil extracellular traps.

**Table 1 nutrients-14-01538-t001:** Summary of current literature on prospective controlled vitamin D supplement trials.

Year	Subjects	Vitamin D Supplementation		Sample Size (F/M)	Age (Range)	Baseline 25[OH]D (ng/mL)	Major Findings	Ref.
		Regimen	Duration					
1991	Children with sustained active polyarticular JIA	25[OH]D (1–2 μg/kg/day)	12 months	13 (12/1)	5–18	28 ± 16	✓ There is a significant improvement in mean 25[OH]D level but not 1,25[OH]_2_D. ✓ Serum PTH decreased at 6 months and persisted to 12 months. ✓ BMD did not change significantly after 6 or 12 months of intervention. ✓ There is no change in disease activity.	[110]
1994	Children with rheumatic disease and osteoporosis, treated with corticosteroid	Ca ± Vit D	6 months	10 (six with JIA)	13 (10.9–18.0)	28.04 ± 8.48	✓ Spinal bone density significantly improved with supplementation. ✓ Mean BMD decreased soon after Vit D and calcium supplementation were terminated.	[109]
2008	Children with oligo-arthritis or polyarthritis	Vit D3 400 IU/day + Ca (1000 mg/day) or Vit D3 (1600 IU/day), or both	6 months	18 (12/6)	10 (5–15)	32.84 ± 15.48	✓ With Vit D3 alone and Vit D3 + Ca supplementation, 25[OH]D levels were increased and 1,25[OH]_2_D levels were unchanged. ✓ Serum Ca was increased in those with Vit D3 and Vit D3 + Ca treatment. ✓ Levels of bone turnover markers and increases in bone mineral content did not differ by treatment.	[90]
2008	Children with JIA free of systemic corticosteroid 3 months prior to enrollment	Vit D3 (400 IU/day) ± Ca (1000 mg/day)	24 months	198 (141/57)	11.7 (6–18)	NA	✓ The average total BMD increased linearly over time in both Vit D3 + Ca and the Vit D3-only group. ✓ BMD increased at a slightly faster rate in the Vit D3 + Ca group than in the Vit D3-only group.	[111]
2019	Children with JIA	±Vit D3 (2000 IU/day)	6 months	36 (13/23)	6.9 ± 3.1	13.316 ± 5.148	✓ Vit D3 supplementation raised serum levels of 25[OH]D in JIA patients. ✓ No reduction in disease activity. ✓ No improvement in BMD.	[108]

## Data Availability

Not applicable.

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
