# Peer review of "Vitamin D Supplementation in Patients with Juvenile Idiopathic Arthritis"

_nutrients, 2022, doi:10.3390/nu14081538_

Round 1

Reviewer 1 Report

Overall, this manuscript is well structured and comprehensive. Some sentences do not appear linguistically correct, so proofreading by a native speaker is recommended. 
From my point of view, only the paragraph under 5.4 seems somewhat unclear (p 8, lines 304 to 309). Here, the results of the cited studies should be presented a little more clearly and in more detail:

"While vita-304 min D status was independent of VDR genotypes, polymorphisms of ApaI and Cdx2 were 305 significantly different between JIA patients and unaffected controls[7, 92]"-> What does that mean in the context of Vitamin D metabolism?

"Recently, the gene–gene interaction between PTPN2 and vitamin D pathway 308 genes was demonstrated and replicated in 2 independent JIA case–control cohorts." -> Again, what does that mean in the context of Vitamin D metabolism?

Author Response

Dear Sir: Please have the attached file for the response to the Reviewers.

Thanks for the help

Best regards,

Jenn-Haung Lai, MD, PhD

Professor

Chang Gung Memorial Hospital

Reviewer 2 Report

Wu et al review data on the mechanisms of VitD for immunology and bone homeostasis in a setting of JIA. I enjoyed reading the review. For me it was a comprehensive read. The figures were very helpful. I am in favor of publication however I do have some recommendations which I feel will improve the manuscript:

- I feel that the immune section is concise, which is good, but some aspects are missing, what about the granulocytes? They are now missing and there is evidence that they do express VDR and react to VitD. Also one study that directly demonstrate a role for VDR in effTreg differentiation in JIA is now missing (PMID 33976194).

-I would appreciate a few more conclusions at the end of every topic. So taken together, what is the overall role on VITD in bone homeostasis or immune modulation? In extension of this, the general discussion could be extended a bit. Where is the field moving? Where are the gaps in knowledge and how can we fill them?

As minor point, the manuscript could benefit from a read through for language errors. For example Significantly>Significant Line 347    missing word line 353-354      dosing>dosages line 370       study>studies line 380   trails>trials line 387      2x been line 395  and others

Reference format is different line89, 221, 308

Author Response

(The authors gave the same response as above.)
